# ‘It’s Hard to Make Good Choices and It Costs More’: Adolescents’ Perception of the External School Food Environment

**DOI:** 10.3390/nu13041043

**Published:** 2021-03-24

**Authors:** Colette Kelly, Mary Callaghan, Saoirse Nic Gabhainn

**Affiliations:** Health Promotion Research Centre, School of Health Sciences, National University of Galway, University Road, H91 TK33 Galway, Ireland; mary.callaghan@nuigalway.ie (M.C.); Saoirse.nicgabhainn@nuigalway.ie (S.N.G.)

**Keywords:** food environments, adolescent, qualitative, schools, school food, food choice

## Abstract

Research on the impact of school and community food environments on adolescent food choice is heavily reliant on objective rather than subjective measures of food outlets around schools and homes. Gaining the perspective of adolescents and how they perceive and use food environments is needed. The aim of this study was to explore adolescent’s perception and use of the food environment surrounding their schools. Purposive sampling was used to recruit schools. Mapping exercises and discussion groups were facilitated with 95 adolescents from six schools. Thematic analysis showed that adolescents are not loyal to particular shops but are attracted to outlets with price discounts, those with ‘deli’ counters and sweets. Cost, convenience and choice are key factors influencing preference for food outlets and foods. Quality, variety and health were important factors for adolescents but these features, especially affordable healthy food, were hard to find. Social factors such as spending time with friends is also an important feature of food environments that deserves further attention. Adolescents’ perceptions of their food environment provide insights into features that can be manipulated to enable healthy choices.

## 1. Introduction

During the last decade, eating out-of-home has become habitual and contributes considerably to dietary habits and nutritional status worldwide [1,2,3]. Food prepared out-of-home tends to be less healthful than food prepared at home and is associated with both increased dietary fat intake and body fatness [4,5,6]. Eating out-of-home is not confined to adults, with adolescents in the US consuming one-third of their energy outside of the home [7]. School and other educational establishments and local food outlets, especially fast food and convenience stores, are common sources of food for adolescents in the US, UK and Ireland [7,8,9,10,11].

The surge in eating out-of-home has seen a parallel increase in research exploring the type and impact of food environments in and around homes, schools and workplaces [4,12,13,14,15]. The food environment is defined as “the collective physical, economic, policy and sociocultural surroundings, opportunities and conditions that influence people’s food and beverage choices and nutritional status” [16]. Different components of the food environment include the variety, quality and price of available foods and also structural aspects, such as the spatial accessibility of retail outlets [17]. Studies of these contextual determinants of diet have predominantly considered associations with food availability measured in terms of proximity to, or density of, retail food outlets or the in-store environment, including the cost and store location of healthy and unhealthy foods [18,19].

The association between characteristics of the food environment, diet and obesity remain equivocal [20], with heterogeneity in approaches to measuring the built environment as just one contributory factor [21]. Subjective measures include perceptions of individuals living or working near food stores, and objective measures are derived from observations, audits or are calculated from existing spatial data using Geographic Information Systems (GIS). Poor or moderate agreement between perceived and objective measures have led to suggestions that these measures should not be seen as comparable [21]. According to Caspi and colleagues [22], perceived measures of the food environment may be more strongly related to dietary behaviours than objective measures and may incorporate dimensions of food access not captured in available objective measures. 

Research on the impact of school and community food environments on adolescent food choice is heavily reliant on objective rather than subjective measures of food outlets around schools or homes. Gaining the perspective of adolescents and how they perceive and use food environments will help us understand how they navigate to and choose particular food outlets. Caraher and colleagues [23], using a mixed-methods approach, found that unhealthy food options were a feature of the streets surrounding schools in a London borough and that cost and attitude to school meals and shop-bought products influenced adolescent choice. These types of data serve to help us with the development and refinement of behavioural interventions addressing food eaten out-of-home, as well as providing estimates of the likely impact of environmental or policy interventions.

The dearth in qualitative research on food environments [24], specifically a lack of subjective measures of adolescent food environments, and existing data demonstrating that close to three-quarters of post-primary schools in Ireland are located within 1 km of fast-food restaurants [25] provided the impetus to explore how youth perceive and use food outlets close to their schools. The current study contributes to the extant literature by exploring young people’s perceptions of external school food environments without restricting the discussion to fast food or takeaway outlets alone. Additionally, this study aimed to explore what barriers and enablers, if any, are at play for young people when accessing food stores. This study focuses on adolescence, a time when young people have greater autonomy, more pocket money and independence travelling to and from school than younger children [9,26]. This study was underpinned by the recognition that children are active social agents and experts in and of their own lives [27] and have a right to participate in research likely to affect them [28]. 

The objectives of the study were to explore adolescents’ perception of the external school food environment, to understand the barriers and facilitators of food outlets and food choice, and whether and what type of changes they would like to the food environment surrounding their schools. 

## 2. Methods and Materials

A qualitative research design was adopted due to the limited knowledge available on adolescent perceptions and experiences of using external school food environments. Objective quantitative measures of food outlets surrounding schools were not collected as our previous work showed that the number of food outlets within a 1 km radius of 63 post-primary schools was an average of 3.89 coffee shops and sandwich bars, 3.65 full-service restaurants, 2.60 Asian and other “ethnic” restaurants, 4.03 fast-food restaurants, 1.95 supermarkets, 6.71 local shops and 0.73 fruit and vegetable retailers of the schools surveyed [25]. 

### 2.1. Sampling and Recruitment

Purposive sampling [29] was used to recruit schools; single and mixed-sex schools located in cities and towns were approached and provided with information about the research. Both younger (first and second year) and older children (Transition Year) were also included in the sampling strategy. Principals/Head Teachers were initially contacted by mail and telephone. If they agreed to participate in the study, researchers provided information sheets and consent forms to be distributed to students and their parents or guardians at home. Information sheets comprised an opt-out or opt-in consent process, depending on the school management’s preference. All students provided informed parental consent and individual assent before participating in the study. 

Multiple methods were used in this study to meet the objectives and were based on participatory mapping techniques [30], photographs used in related school food environment research [31] and group discussions. 

### 2.2. Materials Preparation

The researchers walked the external school food environment prior to the workshops, taking note of the types of food premises within walking distance (1 km) of the school. A smartphone and iPad were used to take photos of the food outlets. 

Three methods were used to collect data: Workshops were conducted with groups of children within the school building (e.g., school hall/auditorium and canteen). Large maps that detailed the location of the school and the surrounding areas were prepared in advance and pinned to the room walls (Figure 1). Working together, the students were invited to place ‘photographs’ of landmarks/buildings onto the map that had been prepared by the researcher. This exercise was used to encourage active participation in the research, to create debate and to enable students to locate themselves and their school within their external school food environment.Each student worked alone on a smaller version of the map described above and wrote the names of food outlets within walking distance of their school onto the maps. Students then highlighted the food outlets they personally use and also drew a route on the map to illustrate how they interact with their own food environment.Facilitated discussions with adolescents were conducted using a semi-structured guide. Students could refer and reflect on the map exercises during the discussions. Students were in discussion groups with peers from their own year group. The age varies within year groups in Ireland (e.g., children in their second year in this study were aged between 13–15 years).

The discussions were recorded, transcribed and analysed using NVivo 12 qualitative data management software (QSR International Pty Ltd., Victoria, Australia), which was also used to manage the data. Thematic analysis was used to analyse the data inductively [32]. Data were initially reviewed for interesting features, with memos being used to record identified themes/patterns. Further analyses focused on more detailed line by line coding followed by interpretation of the meaning of, and relationships between, the initial themes and patterns within and across schools. Differences by school, year group and gender were also explored. The maps were collated and, together with the photos taken by the researchers, were used to visualise the food outlets and to help understand what young people described in the focus groups. These data helped the researchers to visualise the food outlets that young people described in the workshop discussions and provided additional insights into the contextual factors described by students. This also enhances the trustworthiness of the data and findings. Collectively, the authors have backgrounds in nutrition, health promotion and geography. Reflexivity was used to reflect on the role of the researcher(s) and key decisions made.

### 2.3. Ethics

Full ethical approval was obtained from the Institutional Research Ethics committee. All students provided informed parental consent and individual assent before participating in the study.

## 3. Results

In total, 95 students from six schools (one Male, two Mixed and three Female schools (63% girls and 37% boys) located in towns and cities from years 1–2, (12–15 years) and years 4, 5 and Transition Year (15–18 years) took part. Transition Year (TY) is a one-year programme before students enter their Senior Cycle. All data collection sessions lasted a class period (40 min).

The internal food environment differed across schools, some with canteens, vending machines or tuck shops and others with no or very limited facilities. While most students had a choice whether to leave school at lunchtime (four out of six schools, with a fifth school only allowing older year groups to leave), children in one school were required to do so. Most students brought some food (e.g., sandwich/snacks) from home to school each day, especially for ‘small break’ (e.g., sandwich, cereal or chocolate bar), which is typically at 11 am. Packed lunches were also often brought in for the longer lunchtime break.

Almost all students described using retail food outlets at least once a week, with many going 2–3 times a week and some students going to these outlets every day. Students went to the shops before, during and after school, and the vast majority walked at these times. Food was purchased after school if they were hungry, waiting to be collected or had extra-curricular activities such as sports or study periods. The frequency of going to the shops depended on many factors such as whether lunch was brought in from home, the amount of money they had, the weather, for a variety of food or if they wanted something ‘nice’. The perceived number of food outlets within walking distance of schools varied, with between four and fourteen outlets recorded on the maps and between three and four outlets used routinely by students. 

The themes that illustrate the types and use of external school (Sch) food outlets and the barriers and facilitators relative to food outlets are described below. The quotes from children are faithful to their use of language. Parentheses are placed around the word ‘like’, where it is being used as a functional pause (as a comma) in their quotes.

### 3.1. Theme 1: Cost of Food

Discussions about food were synonymous with cost, and students were less likely to use a food outlet if the prices were too high or ‘deals’ on meals were no longer available. “It’s too expensive” (Female, TY, Sch 3); “Y can be really expensive (like) and you’ve to cut costs” (Male, TY, Sch 4). In particular, students were attracted to ‘meal deals’ (e.g., buy one get one-half price, three for €3 or €4 deals), and they sourced the cheapest option across shops. “And everything in XXX is cheap as well so you get better value for money.” (Female, TY, Sch 3); “...because they’re cheap and (like) you get enough bon bons (sweets) that would last the day” (Male, second year, Sch 4); “XXX have cheaper ice pops” (Female, first year, Sch 3).

The cost of food also determined food choice: “In the X you get (like) soup and brown bread for €3 or else you can get (like) a toasted sandwich with that for a fiver, it’s good” (Female, TY, Sch 6), including healthy or less healthy options: “It’s hard to make good choices and it costs more…everything is ‘on offer’ that you really like” (Female, TY, Sch 6); “Yeah it would take a lot of effort (to eat healthy) because (like) if you’re going…you’d have to buy the wraps and then you’d have to go ….buy the other stuff separately” (Female, first year, Sch 6). Getting value for money was an important determinant of the choice of food outlets and of the food itself.

### 3.2. Theme 2: Convenience 

Students favoured food outlets that were convenient such as close to the school, on route to school/home, outlets that provided shelter from the rain, and provided fast service: ”It’s just handy to get there” (Female, second year, Sch 4); “They’re convenient to go to, they’re just quick”; (Male, second year, Sch 2); ”..because they’re (like) near to school and fast” (Female, second year, Sch 5)…”it’s within walking distance…” (Male, TY, Sch 4).

#### 3.2.1. Subtheme: Familiarity

Students tended to use food outlets that they were accustomed to and favoured the shops that had what they liked. “They have, (like) what you want and what you need” (Female, second year, Sch 5); “I guess you just know the place well enough to know what you want in it.” (Female, TY, Sch 3); “You’re not going to go to a place where (like), I’ve never been there before, what if I don’t like it, I don’t have time or you mightn’t even have the money to go get something that you do like.” (Female, TY, Sch 3). 

#### 3.2.2. Subtheme: Time Pressure

Students were under pressure to eat and return to school within the time allocated for lunch, and so they avoided going to food outlets that were further away from the school and take more time to get to: “I wouldn’t go to (like) X because it’s (like) a longer distance, (like) maybe X, you wouldn’t have time to get back to school.” (Male, TY, Sch 4). “Since we only have 40 min for lunch you don’t really have time to be (like) trying new places when you think about it, (like) that’s the kind of thing for the weekend (like), if you know that you’re going to like something in X, you’ve 40 min so you’re going to go and get it there” (Female, TY, Sch 3). 

#### 3.2.3. Subtheme: Too Busy

Some food outlets were very busy, which means the students had to wait longer to be served within the lunchtime limit. “It’s too packed maybe because of all the other students going” (Female, TY, Sch 3); “Well (like) say if there’s loads of people, (like) I could spend my 10 min break to get served” (Female, TY, Sch 4).

### 3.3. Theme 3: Choice

Students described a variety of food outlets and types of food available for purchase. Food outlets close to the school included: large supermarkets, smaller outlets, petrol stations, newsagents/sweet shops/corner shops as well as discount stores. Fast food outlets included chain fast food outlets, Takeaways (e.g., Chinese, Indian, Kebabs, and Mexican), local fast food outlets (“Chippers”) as well as Pizza outlets. Restaurants (i.e., Chinese, Italian, American, and Fish and Chips), hotels, cafes, and pubs were also described. Students felt there were a ‘lot’ of food outlets close to their schools, with some estimating between six and seven stores and others felt there were over 10–11 outlets to choose from. 

Popular outlets ranged from large supermarket chains and smaller chain stores. Deli counters were a particular attraction of these stores, which were also a draw for students in local, independently owned cafes. Larger supermarket stores with fruit and yoghurt as options were mentioned, but shops, whether supermarkets, discount stores or local sweet shops, that sold sweets were very popular. Students were not loyal to any particular shop. 

#### 3.3.1. Subtheme: Food Options

While there were a variety of food outlets, as described above, the foods chosen were similar across students groups. The food frequently described by students were hot chicken fillet rolls (mentioned the most), wedges, chips, pizza and sausage rolls. Other products frequently discussed were sweets, crisps, chicken curries, burgers, Pot Noodles, hot dogs, chocolate, biscuits, cookies, donuts and buns. Other food for sale in the shops included pancakes, crepes, bread rolls with a filling of choice, soup and sandwiches. Drinks commonly consumed included, Energy drinks, Coke, 7-Up, Lucozade, flavoured water, tea, coffee and Miwadi (cordial), which was often offered as a ‘free drink’ with a hot food product. 

Students discussed food and beverage options they perceived as more healthy such as water, milk, fruit, salads, fruit salads, wraps, sandwiches (wraps/toasted sandwiches/paninis), soup, brown bread, lasagne, chicken curry and rice, vol-au-vents, nuts, sesame sticks, popcorn, yoghurts and ‘health’ bars. 

#### 3.3.2. Subtheme: Quality and Variety of Food 

Students made food outlet choices based on the quality of the food on offer; they clearly had preferences for some stores over others because of how fresh, in-date and/or tasty they perceived the food to be. “Well I kind of like going to XXX because it’s kind of ... just everything is tasty stuff because (like) most things over in XXX’s are out of date so.” (Male, second year, Sch 4). “but also (like) I feel like the deli is (like) way fresher than in XXX because I compared kind of, I bought the same thing in X and in Y and it was way better in Y. It was way fresher.” (Female, second year, Sch 4). “The fruit in (like) the corner shops, they’re not as good as in XXX”…(Female, TY, Sch 3). “Yeah (like) the popular things like the bon bons and drinks, they’re not, but (like) the bars, they’re left there for months. If you check the date they’re (like) 4 months out of date”. (Male, second yr, Sch 4). “XXX (chipper) is (like) really greasy” (Male, fourth year, Sch 1) “Some of the little shops are small (like), a little small, so they don’t have (like) that much stuff” (Female, TY, Sch 5). “It’s (like) the same stuff, (like) you kind of get sick of the same stuff all the time” (Female, TY, Sch 5).

#### 3.3.3. Subtheme: Food Environment Is…”More Unhealthy than… Healthy” 

Students had mixed opinions of their food environment, with some being satisfied with what was available in and around their school; “There’s plenty of choices. Yeah, (like) different types of food in places so like fast food or chicken fillet rolls or subs or any of that (like), there’s just a lot of different choices” (Male, TY, Sch 2). Others highlighted a need for a greater selection of food outlets that serve healthier foods. “If you were getting food out, it’s… more unhealthy, than there is healthy.” (Male, fourth year, Sch 1); “Just something (like) more healthy (like) coz everything else is not too great.” (Female, fourth year, Sch 1). Coupled with this was a need for (healthy) food that is more affordable; “The healthier you buy stuff it kind of tends to be more expensive, (like) you’ll go to Xs and get (like) a soup or something. It’ll be really expensive whereas you can get a drink, a roll with chicken in it already for the same price. If not cheaper with a drink.” (Male, second year, Sch 2). 

### 3.4. Theme 4: Social Relationships

Food was not the only reason for going to food outlets; at times it was to buy supplies for school, but invariably it was because they wanted to go for a walk or to spend time with their friends who were going, and to meet others; “A lot of people just stand outside X rather than go in and buy something” (Male, fourth year, Sch 1); “Sometimes you don’t (buy anything). Sometimes (like) your friends would be buying so then you’d just wait with them or something” (Male, TY, Sch 2); “Your friends want to go”, (Female, fourth year, Sch 1). “Well mostly (like) people are going there anyway so you go…” (Male, second year, Sch 2).

#### 3.4.1. Subtheme: Food Outlet Staff

Students avoided food outlets if staff were unfriendly, or if rules are in place that will require them to leave the premises if they do not buy food. “XXX you’re only allowed in if you’re buying something, they’re really strict about that” (Female, first year, Sch 6). “I just don’t like the shop….because the shopkeepers are mean” (Female, first year, Sch 3).

#### 3.4.2. Subtheme: Space and Seating

Some outlets are deemed to be too small by the students, with no place to sit and eat comfortably; “… so (like) you’ve time to eat it (like) and sit down and stuff.” (Female, TY, Sch 4); “It’s easier (like), you go to XXX you can come down and actually sit down and eat it rather than walking and eating at the same time” (Female, second year, Sch 4); “Yeah and I like it because …and you have a place to sit there so” (Female, TY, Sch 4).

Differences between schools were only evident for the Social Relationship theme, whereby students in schools that could not leave at lunchtime did not describe data relevant to this theme. Because of the age range in year groups, differences by age were not discernible from the data collected, and analysis by year group did not illustrate clear differences between younger and older year groups. Gender differences were found for the Social Relationships theme whereby only female students reported food outlet staff and space and seating as important factors influencing their choice of food outlet. All of the remaining themes were identified from data collected across school types, year groups and gender. 

## 4. Discussion

The availability, type and nutritional quality of food available in and around post-primary schools have been well documented in Ireland [9,25,33,34], yet this is the first study to explore adolescents’ perception of their external school food environment. Qualitative data from 95 young people in six schools were collected using mapping tool exercises and discussion groups to capture their perception of the school food environment.

Students describe a variety of food outlets surrounding post-primary schools; this supports existing data from both quantitative and qualitative research [9,25,33]. There is a range of large and small outlets, and adolescents are not loyal to particular shops. However, the appeal of outlets with ‘deli’ counters was clear and especially those with hot food, such as those that sell hot chicken fillet rolls. This is similar to other studies of local retail outlets in Ireland [9] and in the UK [35]. Purchases from corner shops in the US illustrate that beverages are the most popular item among adolescents, although this data involved younger children than our study and included purchases over the year, not just during the school term [36]. The popularity of sweets was also evident in our study and is supported by other work in Ireland [9] and the US [36]. Studies that have captured the density of fast food outlets close to schools are likely to miss the availability of fast food in other outlets, be that small corner stores or supermarket chains. Similarly, the availability of sweets in a variety of stores in our study questions the protocol of categorising supermarkets as ‘healthy’ and fast food outlets as ‘unhealthy’ in food environment studies [10,37,38,39]. 

Factors can act as both facilitators and barriers, and this is evident in our data related to adolescent food outlet use and food choice. Factors such as proximity and convenience, time pressures and familiarity are difficult to change and would require changes to school timetables to enable longer lunch breaks, giving flexibility for students to explore other shops and food outlet options and to sit and eat. Factors more amenable to change include the cost of food, and given the explicit desire to get value for money, pricing and food marketing are likely to be important intervention functions. The influence of marketing (i.e., in-store meal deals, free drink options) on the choice of food outlets and foods themselves was also evident and is related to adolescents getting value for money. Getting value for money was important for youth from low socioeconomic status (SES) backgrounds but not mixed SES backgrounds in the UK [31]. These attributes of sourcing better value for money and awareness of price and promotion should be valued among adolescents and are likely to reflect what educators teach, and families discuss. Of particular interest is how youth appreciate and seek out outlets that regularly change their menu options, providing them with a variety of food options. Quality produce also featured heavily in the discussions, with adolescents citing fresh produce and taste as factors that determine their choice of outlet and food. Taste has long been a clear determinant of food choice, but the emphasis on quality and variety as important factors for adolescents is not clear in the literature. 

A deterrent for some students relates to the internal architecture of the food outlets, with a lack of seating and limited space an obstacle to frequenting certain food outlets. This has also been described as a barrier to the use of school food canteens [31,34] and highlights the importance of physical factors on adolescents’ use of food outlets and, consequently, their related food behaviours. In addition, the rules imposed by staff and the unkind attitude of staff to students are also barriers to the use of some food outlets. This may come from a generally suspicious attitude towards teenagers, which has also been reported elsewhere [31]. Interestingly, shop staff in the latter study treated young people from low SES areas with respect, valuing their custom. The influence of peers was noted in terms of going to shops with friends, but peer/peer group influence on food choice within stores was not explicit in our study. Choosing and eating food are shared activities through which young people can express affinity to preferred peer groups and thus ‘fit in’ [40,41,42]. Our data illustrate that both the physical food environment and the social aspect of food, and how these interact, play a role in how adolescents navigate their external school food environments. Indeed the social aspect of food is thought to need more emphasis when exploring and intervening in adolescent food environments [43]. 

The data from this study clearly shows that teenagers are primarily driven to certain outlets by the price of food. While fast foods and sweet snacks were popular among adolescents, healthier options equated to higher cost, perhaps because the ‘offers’ or ‘deals’ were on high-fat, high-sugar foods and non-perishable goods. Within schools, the mean price of healthy lunch items has also been shown to be greater than less healthy items, highlighting an opportunity for pricing strategies both within [12] and outside schools. While some were content with their current food options, others wanted to see a change, with more variety and more healthy options at affordable prices a requisite. A range of interventions that manipulate price, suggest swaps, and manipulate item availability in shops/grocery stores have examined changes in food purchasing behaviours [44] or health-related outcomes [45], albeit among adults or families [46]. Little is known about the impact of store interventions on adolescent food purchasing behaviour. Our study can help us understand why adolescents use certain outlets/choose certain foods and their interest in changes to the food environment; adolescent’s attitude to change has also been lacking in related interventions in school settings [47] even though the effect of interventions is highly influenced by people’s attitudes towards them [48]. There have been calls for the involvement of youth in the design and implementation of school food changes [33,34,49], with some progress in their involvement in policy development related to food environments [50]. Interventions to improve food choice options in stores and that target food purchasing behaviours (e.g., marketing) should also involve youth to effect behaviour change. 

This is the only study focused specifically on Irish school children’s perception of the local school food environment; one other study focused on stakeholders (staff and students) perception of school food generally [34]. It also incorporated the use of maps with youth, so they could work individually and collectively, which enabled debate but also provided an opportunity for each child to contribute. The maps and photos provided a perspective of the food environment close to schools, while the group discussions provided insight into their reality. A strength of this study is reporting in line with the Standards for Reporting Qualitative Research guidance [51].

However, there are some limitations to the study, including that data on the availability and cost of school meals (subsidised or not) or the average spend on food inside and outside of school were not collected. Similarly, data on whether subsidised school meals were available in schools was not collected. Our data cannot provide insight into whether food options within schools influence food choice outside of school. The food outlets mapped by participants were not checked for accuracy, nor was the distance travelled by students. School policies differ across Ireland on whether children can remain or leave school at lunchtime. In our study, just two groups of children, out of the twelve discussion groups, were not allowed to leave school at lunchtime. Indeed, some schools, often due to lack of catering facilities and limited space, mandate (older) children to leave the premises at lunchtime. All of these issues pose challenges for those offering guidance, toolkits or implementing policies on healthy eating for schools [34]. Indeed school food policies and practices that are limited to foods sold at school can only go so far in influencing the dietary behaviours of students [9,52], and changes to the external school food environment are also needed to enable healthy food choices. 

Adolescents frequent the external school food environment regularly, and the food choices made can contribute to nutritional habits and nutritional status. Understanding the drivers and facilitators of food outlets and food choice for adolescents is key to developing effective and sustainable interventions. Our study provides insight into the perception and use of food environments around schools and gives a nuanced understanding of the food environment from young people’s perspectives. The human perception of environments should be considered [53] when exploring related behaviour or changes to the local environment, and understanding young people’s perspectives should also be valued [54]. Our data also supports the value of qualitative studies of the food environment to generate a comprehensive picture of food availability and food choice.

## Figures and Tables

**Figure 1 nutrients-13-01043-f001:**
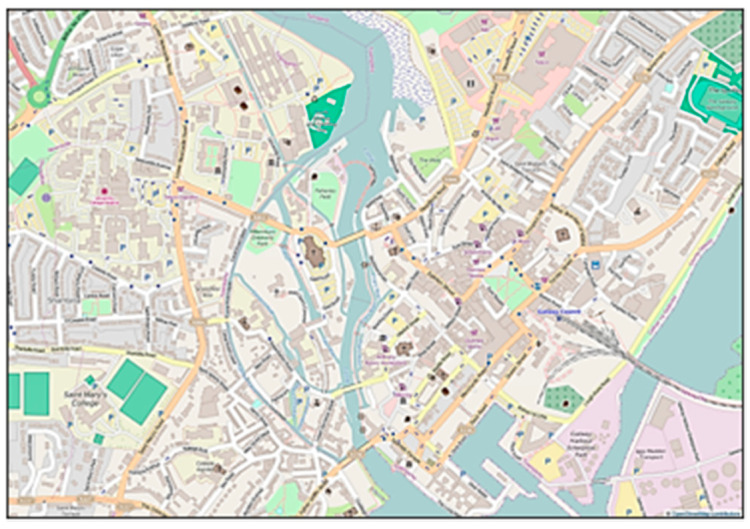
Example of a large map used during workshops with adolescents (© OpenStreetMap contributions).

## Data Availability

The data presented in this study are not publicly available in accordance with the type of consent obtained about the use of confidential data.

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
