# Peer review of "‘It’s Hard to Make Good Choices and It Costs More’: Adolescents’ Perception of the External School Food Environment"

_nutrients, 2021, doi:10.3390/nu13041043_

Round 1

Reviewer 1 Report

Thank you for a very interesting manuscript exploring the important issue of school food environments utilising a qualitative, 'lived experience', methodology.

The introduction provides a strong rationale for the importance and purpose of this research.

I would recommend that you use the SRQR or COREQ reporting guidelines to ensure inclusion of important information pertaining to the reflectivity of the researchers, theoretical underpinnings of the study design, etc in the context of qualitative research. 

Please check consistency of the use of italics for quotations in the results.

The study identified interesting themes, although the way in which these are presented appears to follow more of a descriptive coding structure ('bucket themes') as opposed to a thematic analysis. The results are describing types of outlets, enablers (drivers) and barriers - these are 'bucket' themes or domain summaries.  The next step is to interpret the patterns/connections between these to arrive at few central themes - which in this instance appear to be cost, convenience (time, proximity, familiarity, too busy), social (space, seating, staff), and choice (quality, variety, preferences, food options).

The above suggested re-organisation would also link in with your existing discussion which describes intervention strategies addressing these main themes.

A strength of this study is the participatory action methodology used, involving the end-user, in this instance, adolescents, in data collection and interpretation, and identification of potential solutions (intervention strategies). 

Two other references you may find of use regarding the need for caution when using tools to measure food environments: 

https://www.ncbi.nlm.nih.gov/pmc/articles/PMC6651399/

https://ijbnpa.biomedcentral.com/articles/10.1186/s12966-020-01019-1

Reviewer 2 Report

This is an interesting approach to determining the key driving factors behind school childrens use of food outlets and is well written. It benefits from showing the impact of all available food outlets surrounding schools and does not just focus on fast food stores. While the information provided was interesting, there were several areas that I would have liked more information about;

Methods

  • Did children complete the study groups with friends? How did this influence their responses.
  • How many children were in each discussion group? Were they in mixed age groups?
  • What is the average price of a school meal? What was the typical spend when children eat out of school?
  • Were any of the children receiving free school meals? If so, did they still choose to eat out?

Results

  • There is no information of sex or age differences in the thematic responses.
  • There was mention that some of the children had no choice but to eat out of the school every day. What proportion of kids that had a choice of in and out of school facilities? Did these children forced to leave school have different criteria to those that had a choice of in and out of school lunches?
  • Ln 143 implies that younger children were more likely to eat in school while older ones ate out. Is this the case?
  • How did the range of choice within the school affect the types of food outlets preferred out of the school?
  • Can more be made of the individual map information. What are the food outlet densities about the schools? Can you tell from the maps the typical distance travelled was for different age groups?
  • Was there an age or sex difference on perception or importance of healthiness of the choices?

Minor comments

Some of the language is specific to the study location and may be unclear to others. What are bon bons for example and Miwadi?

When direct quotes from the children are included, can the language be edited to be more familiar to a non-native English speaker. The repeated use of the word “like” for example is very confusing.

Ln 166 spelling of because

Italics for quotes missing from ln 205-217 and ln 230-233.

Ln 293 This study does not provide data on the amount of food outlets surrounding the schools, only the childrens perception of the number.

Reviewer 3 Report

Background:

It is unclear that the following sentence means:

“Overall, it seems that technology does not always represent reality and that perception may be a mediator between objectively measured exposure to environments and interaction [23].”

Sampling and recruitment

Could you elaborate a bit more what the purposive sampling entailed?

Students also provided written consent. Should you include “Parental” before consent?

Results

The section on “price determines food choice” needs more work. How is it different from the value from money theme? In addition, the first two paragraphs have nothing to do with food price. You mention: “This theme describes food discussed by the adolescents but the main driver was price.” That is not convincing

Theme 6: More than just the food …the social aspect is important The first several sentences under this section again have nothing to do with the social aspect of the food environment. You have text covering several issues. This section also needs rewriting

Discussion

Remove the links in the discussion and instead add those websites as references.

“However, there are some limitations to the study including that 369 the Irish school context is likely to differ from other countries, although there is little uniformity across countries in any case [12].” This is not a limitation of the study; remove

Round 2

Reviewer 2 Report

The authors have addressed the majority of the comments satisfactorily. 

There are 2 areas that could have been explored further but have not been fully addressed. 

1. Map Data: The methods state that the area surrounding the school was explored with photos taken of all food outlets. This means that you can accurately calculate exactly what proportion of food outlet were identified by the children from each school. Once the children identified a specific outlet in their mapping that they visited, It must therefore be possible to determine distance from the outlets to the school. Your response stated "Distance travelled could not be calculated accurately since the students did not plot exact locations on the map." Even if incorrectly mapped by the children, you can calculate the correct walking distance if the establishment has been named.

2. Age/sex differences. I feel that this is important to add and would enhance the manuscript. 

You state that "We did not set out to look at differences by year group, age or sex." but that does not mean that it would not be interesting to do so. You also state that it wasnt possible as some groups were mixed sex and mixed age. There is information in the manuscript that the discussion groups were separated into 2 age groups so it is unclear why this was not possible or desirable to explore these separately. I would imagine that by knowing who was in the study groups, the sex of the person speaking would be easily identified by the interviewer even in a mixed group. The quotes actually state the sex of the person and therefore this must already have been done. 

There is some conflict in the methods and results as methods state that yr 2 are 13-15 year olds while results state that one discussion group was year 1,2 (aged 12-14) while the other was year 3, 4 and transition (aged 15-18).
